# Influence of the Pilot Flame on the Morphology and Exhaust Emissions of NH₃-CH₄-Air Swirl Flames Using a Reduced-Scale Burner at Atmospheric Pressure

Cristian D. Avila Jimenez [1,2,*], Santiago Cardona [1], Mohammed A. Juaied [1,2], Mourad Younes [3], Aqil Jamal [3], Thibault F. Guiberti [1,2] and William L. Roberts [1,2]

1    Clean Combustion Research Center, King Abdullah University of Science and Technology (KAUST), Thuwal 23955-6900, Saudi Arabia
2    Mechanical Engineering Program, Physical Science and Engineering Division, King Abdullah University of Science and Technology (KAUST), Thuwal 23955-6900, Saudi Arabia
3    Saudi Aramco, Dhahran 31311, Saudi Arabia
*    Correspondence: cristian.avilajimenez@kaust.edu.sa; Tel.: +966-0543457590

**Abstract:** This work presents an experimental study on the influence of the pilot flame characteristics on the flame morphology and exhaust emissions of a turbulent swirling flame. A reduced-scale burner, inspired by that fitted in the AE-T100 micro gas turbine, was employed as the experimental platform to evaluate methane ($CH_4$) and an ammonia-methane fuel blend with an ammonia ($NH_3$) volume fraction of 0.7. The power ratio (PR) between the pilot flame and the main flame and the fuel composition of the pilot flame was investigated. The pilot power ratio was varied from 0 to 20% for both fuel compositions tested. The $NH_3$ volume fraction in the pilot flame ranged from pure $CH_4$ to pure $NH_3$ through various $NH_3$–$CH_4$ blends. Flame images and exhaust emissions, namely $CO_2$, CO, NO, and $N_2O$ were recorded. It was found that increasing the pilot power ratio produces more stable flames and influences most of the exhaust emissions measured. The $CO_2$ concentration in the exhaust gases was roughly constant for $CH_4$-air or $NH_3$–$CH_4$–air flames. In addition, a $CO_2$ concentration reduction of about 45% was achieved for $X_{NH3} = 0.70$ compared with pure $CH_4$, while still producing stable flames as long as PR ≥ 5%. The pilot power ratio was found to have a higher relative impact on NO emissions for $CH_4$ than for $NH_3$–$CH_4$, with measured exhaust NO percentage increments of about 276% and 11%, respectively. The $N_2O$ concentration was constant for all pilot power ratios for $CH_4$ but it decreased when the pilot power ratio increased for $NH_3$–$CH_4$. The pilot fuel composition highly affected the NO and $N_2O$ emissions. Pure $CH_4$ pilot flames and higher power ratios produced higher NO emissions. Conversely, the NO concentration was roughly constant for pure $NH_3$ pilot flames, regardless of the pilot power ratio. Qualitative OH-PLIF images were recorded to further investigate these trends. Results showed that the pilot power ratio and the pilot fuel composition modified the flame morphology and the OH concentration, which both influence NO emissions.

**Keywords:** ammonia; methane; pilot flame; flame morphology; exhaust emissions; OH-PLIF

## 1. Introduction

Ammonia ($NH_3$) can be directly used as a carbon-free fuel for gas turbines [1–4]. Experiments using $NH_3$-based fuel blends on micro gas turbines (mGT) have been carried out to understand their performance, combustion efficiency, and exhaust emissions [3–6]. However, the major barriers to burning $NH_3$ are the high $NO_x$ emission [7–9] and its low reactivity, making it challenging to stabilize $NH_3$ flames in practical applications. Some burners feature a pilot flame to ease flame stabilization by providing continuous heat and radicals as ignition sources and, in turn, preventing the main lean flame from extinguishing [7,8].

Piloted combustors, such as the one fitted in the Ansaldo Energia AE-T100 mGT, can feature extended lean blow-off limits compared to non-piloted combustors [9]. Thus, a pilot flame could be implemented to stabilize the main flame of, less reactive, $NH_3$-based fuel blends. However, Zanger et al. [9] found that the pilot flame is the main $NO_x$ source from a burner similar to the T100 burner when fueled by conventional hydrocarbons. In addition, Cadorin et al. [10] numerically demonstrated that, for the T100 burner and methane ($CH_4$), operating the burner with 15% of the fuel injected through the pilot produces higher temperatures in the non-premixed combustion zone compared to the operation with 13%, consequently increasing $NO_x$ emission.

One way to stabilize low-reactive fuel blends is to inject more reactive fuels into the pilot flame [11]. For example, a pure $CH_4$ pilot could be preferable to stabilize $NH_3$-air lean flames. However, Kristensen et al. [12] found that HCN oxidation leads to high $N_2O$ production in the burnout zone for blends of CO, NO, HCN, $NH_3$, and $O_2$. Takagi et al. [13] suggested that flames of $NH_3$ blended with hydrocarbons have a higher tendency to produce $NO_x$ than pure $NH_3$ flames because of the higher HCN concentration. In addition, Wargadalam et al. [14] demonstrated that NO and $N_2O$ formation is promoted by HCN oxidation. $N_2O$ is a critical pollutant for $NH_3$ combustion since it has a Global Warming Potential of around 270 times that of $CO_2$ [15]. As a result, $NH_3$-based flames with an $N_2O$ concentration in the exhaust of around 240 ppm can be as harmful as a lean $CH_4$-air flame [16] as far as global warming is concerned.

In addition, the production of NO is mainly determined by the concentration of OH radicals in flames containing $NH_3$ [17] because this radical plays a key role in the NO production via HNO [18]. Okafor et al. [19] found that the larger pool of O/H radicals in the $NH_3$-$CH_4$-air flames with more than a 10% $NH_3$ volume fraction in the fuel blend may produce more NO emissions than $NH_3$-air flames. According to Somarathne et al. [17], the OH concentration in non-premixed $CH_4$-air flames was more than twice that of $NH_3$-air flames. Therefore, measuring the OH pool concentration while varying either the pilot power ratio (PR) or the pilot fuel composition (pilot $X_{NH3}$) could bring insights into NO emissions control for $NH_3$-$CH_4$-air fired gas turbines.

The influence of PR and pilot $X_{NH3}$ on the flame morphology and exhaust emissions, especially NO and $N_2O$, has not yet been reported for piloted burners relevant to mGTs while running on $NH_3$-containing fuel blends. This paucity is what this study intends to address. An optically accessible reduced-scale burner inspired by the one fitted in the AE-T100 mGT has been implemented as the experimental platform. Flame images, CO, NO, and $N_2O$ emissions were recorded while varying the PR and the pilot $X_{NH3}$. In addition, OH planar laser-induced fluorescence (PLIF) images were recorded to further investigate the impact of the pilot characteristics on the flame morphology and NO emission.

## 2. Experimental Setup and Methods

### 2.1. Reduced-Scale Burner and Operating Conditions

Figure 1 shows a schematic of the reduced-scale burner implemented in this study, consisting of a pilot combustion chamber, the main swirl mixing chamber, a secondary air swirler, and the main combustion chamber. The reduced-scale burner features have been described in previous work [20], and it is inspired by the AE-T100 micro gas turbine burner, which has been well described in others' work [21–24]. In the reduced-scale burner, the fuel and air are injected separately into the center of the pilot liner to create a non-premixed swirl flame. This pilot flame produces heat and radicals to stabilize the main combustion zone close to the pilot's liner outlet. The main fuel and air mix, upstream of the main combustion chamber, within the main swirl mixing chamber and exit it with a swirl-like motion through the co-axial annulus surrounding the pilot liner. The proportion of fuel supplied to the pilot and main fuel lines can be adjusted to vary the power delivered by each flame. Finally, the secondary air swirler enhances the fuel-air mixing inside the combustion chamber and prevents flashback. Since the distribution of air is fixed by pressure drops in

the actual mGT burner, mass flow controllers (MFC) were implemented to prescribe the same air distribution.

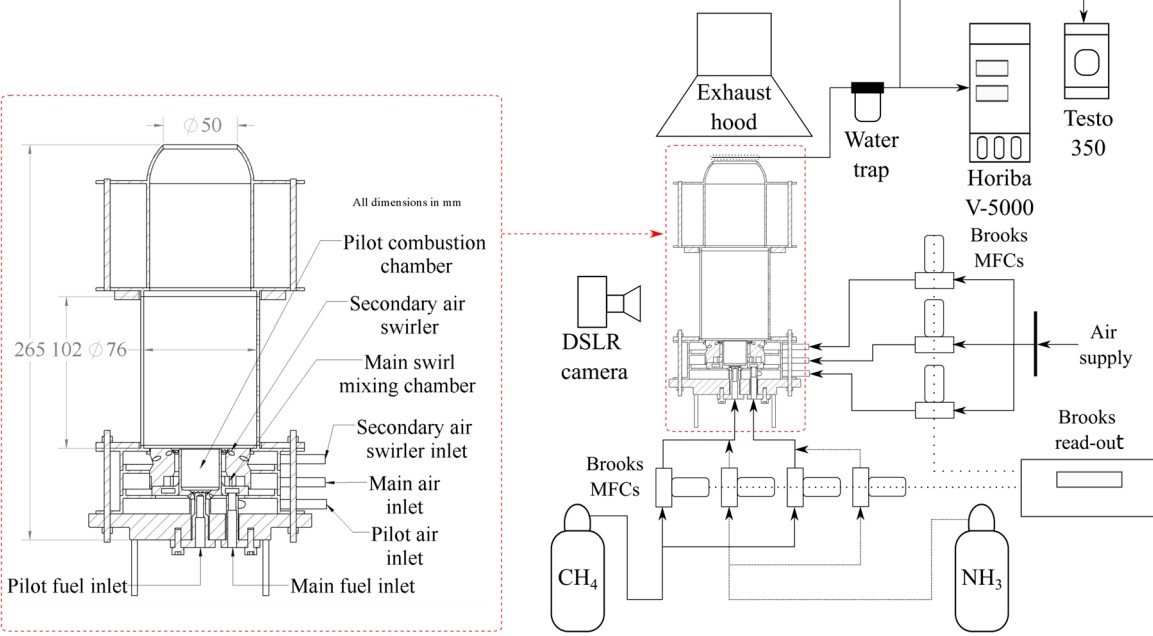

**Figure 1.** Schematic of the reduced-scale burner and experimental setup.

Two series of experiments were conducted to assess the impact of the pilot flame on exhaust emissions and flame morphology. In the first series, the pilot flame's power was varied. In the second series, the fuel composition of the pilot flame was varied. For consistency, the total thermal power was kept constant at 5 kW, and all experiments were performed at atmospheric pressure. Khateeb et al. [25] found that NO concentration peaks at an $NH_3$ volume fraction ($X_{NH3}$) in the fuel blend of around 0.50 and for lean equivalence ratios ($\phi \sim 0.80$) for $NH_3$-$CH_4$-air combustion in a generic swirl burner at atmospheric pressure. In addition, Khateeb et al. stated that NO could potentially be reduced for higher $NH_3$ volume fractions and leaner equivalence ratios if the flame could be stabilized. This was verified by the authors in the present reduced-scale burner and at elevated pressure for $NH_3$-$CH_4$ fuel blends [20]. Flames with $X_{NH3}$ = 0.7 presented similar flame stability compared with $CH_4$-air flames and yielded competitively low NO emissions at an overall equivalence ratio of $\phi_{overall}$ = 0.60, which is more representative of the actual operation of a gas turbine. Therefore, $NH_3$-$CH_4$-air flames with $X_{NH3}$ = 0.70 were investigated at an overall equivalence ratio of $\phi_{overall}$ = 0.60, and $CH_4$-air flames were also tested as the baseline.

The pilot power ratio (PR) is defined as the thermal power of the pilot flame divided by the total thermal power. The PR was varied from 0 to 20% by increments of 5% for both $CH_4$-air and $NH_3$-$CH_4$-air flames, a range that comprises the normal AE-T100 operating envelope [22,23]. PR = 0% means that all the power is delivered by the technically-premixed main flame, while 20% means that the non-premixed pilot flame delivers 20% of the total thermal power. This was achieved by simultaneously modifying the proportions of fuel fed through the main and through the pilot lines, consistent with the actual AE-T100 mGT burner operation.

The pilot fuel composition, indicated by the pilot $NH_3$ volume fraction (Pilot $X_{NH3}$), was varied from pure $CH_4$ (Pilot $X_{NH3}$ = 0.0) to pure $NH_3$ (Pilot $X_{NH3}$ = 1.0). Effects of the pilot fuel composition were tested at three different PRs (5, 10, and 15%). The main flame fuel composition was kept constant, with $X_{NH3}$ = 0.70. Table 1 summarizes the test conditions for the pilot fuel composition test series.

**Table 1.** Pilot $X_{NH3}$ test conditions.

| PR [%] | Pilot Flame Power [kW] | Main Flame Power [kW] | Main $X_{NH3}$ Investigated [-] | Pilot $X_{NH3}$ Investigated [-] |
|---|---|---|---|---|
| 5 | 0.25 | 4.75 | | |
| 10 | 0.50 | 4.50 | 0.70 | 0.00, 0.50:0.10:1.00 |
| 15 | 0.75 | 4.25 | | |

Thermal mass flow controllers (BROOKS Instrument, SLA series) were used to prescribe the fuel and air flow rates. These were calibrated before the experiments with a gas flow calibrator (MesaLabs ML-1020), leading to an accuracy better than 1% for each of the flow rates.

### 2.2. Flame Imaging and Gas Analyzers

The equipment used for flame imaging and exhaust gas analysis is also featured in Figure 1. Flame images and exhaust emissions were recorded each time the PR or the Pilot $X_{NH3}$ were varied. Time-averaged broadband flame images were recorded with a DSLR camera with an exposure time of 5 s, an aperture of $f/4$, and an ISO number of 800.

A stainless-steel sampling probe was installed downstream of the burner's outlet and was designed following [26] for spatially-averaged emission measurement. The sampling gases were conducted to a Testo-350 exhaust gas analyzer to measure $CO_2$, CO, and NO mole fraction and to a Horiba VA-3000 sampling conditioning unit connected to a Horiba VA-5000 to measure the mole fraction of $N_2O$ and $O_2$. A water trap was installed between the sampling line and the gas analyzers to protect them from the high-water content in the exhaust gases. The gas analyzers were calibrated to a precision of 3% and 2%, respectively. Furthermore, exhaust emissions were corrected for an industrial standard of 15% dry $O_2$ [27] mole fraction to compare cases with different conditions.

### 2.3. OH-PLIF

An OH planar laser-induced fluorescence (OH-PLIF) system was used to investigate further the impact of the pilot flame on the flame morphology and NO emissions. The OH-PLIF system consists of an Nd:YAG laser (Continuum, Powerlite DSL9010) pumping a tunable dye laser (Continuum ND6000, dye Rhodamine 6G + UVT) with a 7-ns pulse duration at a repetition rate of 10 Hz. The dye laser was tuned to the OH radical's Q1(6) transition at 282.928 nm and generated an 8 mJ/pulse. A cylindrical concave lens was used to convert the laser beam into a laser sheet, and a spherical lens was used to collimate the laser sheet into a plane with a 50-mm height and a thickness of around 400 μm. The laser plane was vertically aligned to the burner's central axis, as illustrated in the top view of Figure 2.

The laser-induced OH fluorescence near 310 nm was recorded perpendicular to the laser sheet by an intensified CCD camera system (Princeton Instruments, PI-MAX 3) equipped with a UV-lens (LaVision, $f/2.8$, $f$ = 100 mm) and a 20-nm narrow bandpass, high-transmission OH filter centered at 320 nm. The image intensifier gate width and gain were set at 300 ns and 60%, respectively.

Six hundred images were recorded for each operating condition to achieve statistical convergence of the mean values. Background subtraction was applied to each instantaneous image. An interrogation area of 370 × 400 pixels was chosen to calculate the normalized OH intensity close to the pilot outlet. Figure 3 presents an example of single-shot OH image and highlights the selected interrogation area as a red dashed rectangle. The average OH-PLIF intensity was calculated for each image in this interrogation area. Then, the corresponding 600 values were averaged for each condition and normalized by the maximum value found across all operating conditions. In addition, the 600 single-shot images were averaged for each condition and then normalized with the maximum and minimum values found across

all operating conditions to obtain a normalized, time-averaged OH-PLIF image for each condition.

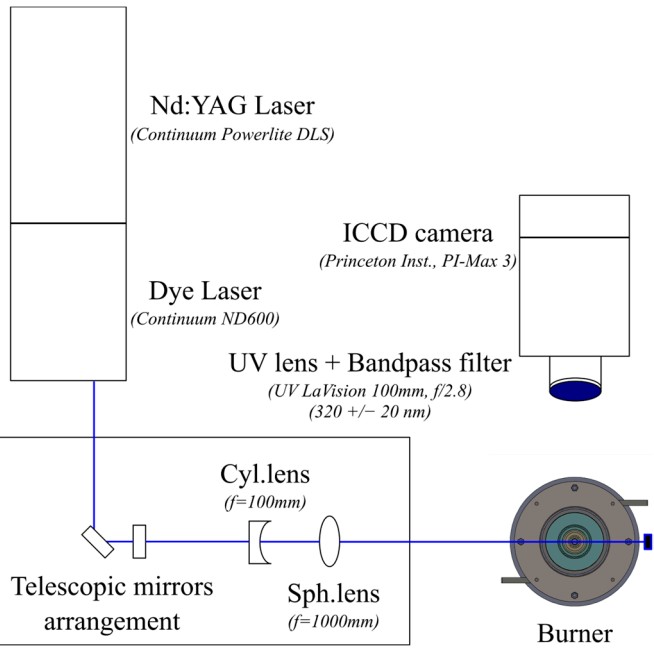

**Figure 2.** Schematic of the OH-PLIF setup.

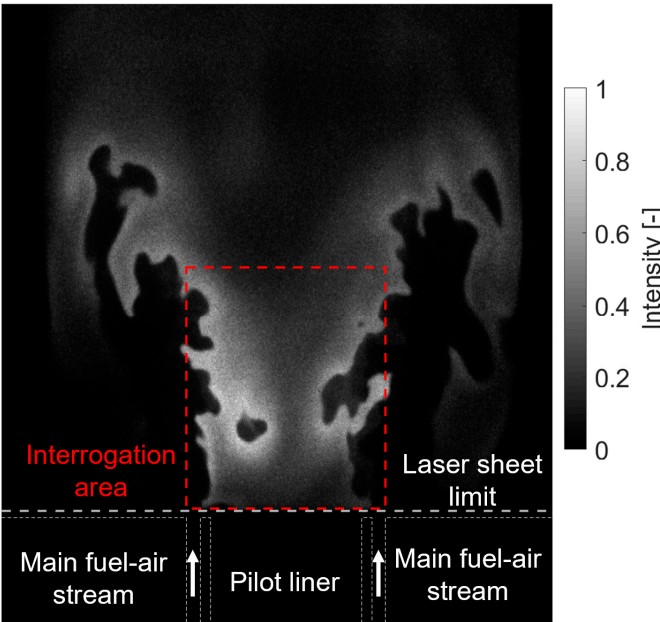

**Figure 3.** An example of a single-shot OH-PLIF image with the interrogation area shown as a red dashed red rectangle.

## 3. Results and Discussions

### 3.1. Influence of the Pilot Power Ratio (PR)

3.1.1. Flame Morphology

Figure 4 shows the time-averaged broadband image (left) and a representative single-shot OH-PLIF image (right) of $CH_4$-air and $NH_3$–$CH_4$–air flames for PR ranging from 0 to 20%. The flames generally present the typical V-shape observed for swirling turbulent flames. By looking at the time-averaged flame images, the $CH_4$-air flame at PR = 0% is

slightly lifted from the pilot liner. The cold air issuing from the non-reacting pilot does not provide support to the main flame, and it probably even induces heat loss. Introducing fuel into the pilot with a PR of 5% or 10% reduces the lift-off height for $CH_4$-air flames, producing a compact flame anchored at the pilot's liner rim. Beyond the 10% threshold, further increasing the PR produces lifted but compact flames. In this case, lift-off may be caused by the larger velocity of expanding hot products issued from the pilot. The large impact of PR on the main flame's structure can also be seen by examining a few representative single-shot OH-PLIF images. Only the flame with PR = 10% features bright and sharp OH layers in the immediate wake of the pilot layer, indicative of a reaction zone.

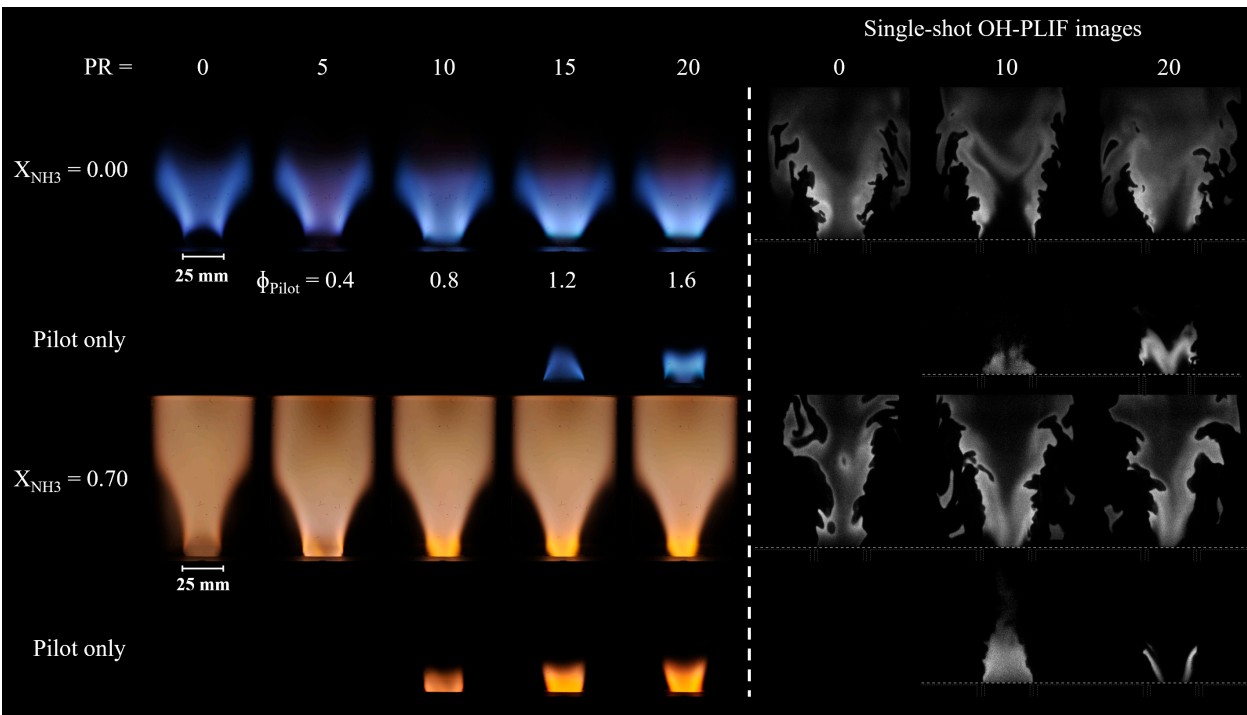

**Figure 4.** Time-averaged broadband (**left**) and OH-PLIF (**right**) images of $CH_4$-air and $NH_3$–$CH_4$–air flames as a function of the pilot power ratio.

Figure 4 also shows images recorded without any fuel supplied to the main flame, i.e., only contributions from the pilot flame can be observed. No visible chemiluminescence is seen outside of the pilot liner for PR up to 10%, but hot burned products are visible from PLIF images for PR = 10%. This implies that conditions are not met for the pilot flame to burn for PR = 5% and that the pilot flame is shorter than the pilot liner length for PR = 10%. Direct broadband and OH-PLIF images show that some of the pilot fuel is burnt outside of the pilot liner if PR > 10%, at least if the main flame's influence is omitted. In summary, Figure 4 shows that modifying PR does not only modify the amount of heat provided by the pilot flame, but it also changes its structure.

The two lowest rows of Figure 4 also show the $NH_3$–$CH_4$–air flames exhibiting the typical orange hue attributed to the $NH_2$ alpha band chemiluminescence [28] and $NO_2^*$ [29]. Longer flames are observed during the experiments compared with $CH_4$-air flames, and none of the flames seem lifted if only direct broadband images are considered. However, a careful study of instantaneous OH-PLIF images shows that there is no thin reaction layer anchored to the rim of the pilot liner, regardless of PR. At this location, only hot burnt products exist, within which OH radicals remain even though no heat is being released, and these are also responsible for the bright orange light emissions immediately downstream of the pilot. The presence of radiating hot products immediately downstream of the pilot liner for PR = 0% is most probably due to a recirculation from the downstream region because of the swirl motion.

Although the flame at PR = 0% does not blow off, an unstable behavior is observed with an intermittent flame lift-off further down the main flame liner and subsequent "reattachment" closer to the pilot liner. Increasing the PR results in more stable and slightly more compact $NH_3$–$CH_4$–air flames, with a more intense orange hue issuing from the pilot. The strong pilot flames shown in the lowest row of Figure 4 for PR > 5% explain this. The instantaneous OH image illustrates that the flame at PR = 0% is elongated and thin compared to the other $NH_3$–$CH_4$–air flames. Notably, the most compact flames were obtained at a PR of 10 and ≥5% for $CH_4$–air and $NH_3$–$CH_4$–air, respectively, suggesting that the optimal PR varies with fuel compositions, if only flame stability is considered.

### 3.1.2. Exhaust Emissions

Figure 5 shows the measured exhaust CO concentration as a function of the PR. It is below 10 ppmvd for all $CH_4$-air or $NH_3$-$CH_4$-air flames. For $CH_4$-air, the CO concentration somewhat increases when PR increases, which is consistent with findings of [10,30] and can be explained by the fact that the non-premixed pilot flame is more prone to yield CO that the lean premixed main flame. However, it can be argued that CO variations are small and within measurement uncertainties and are, as such, not very meaningful. For $NH_3$–$CH_4$–air flames, the maximum CO concentration is found for PR = 0 and it is higher than for the $CH_4$-air counterpart. This can be explained by the unstable behavior of this flame that promotes incomplete oxidation. The minimum CO concentration is found for PR = 5% and is below the detection limit of our measurement unit. This is because this flame is stable, it has a relatively small amount of carbon in its fuel blend, and it has a comparatively weak contribution from the non-premixed pilot flame. Because the pilot flame is more prone to produce CO that the main lean premixed flame, the CO concentration increases when PR is increased beyond 5% also for the $NH_3$–$CH_4$–air flames.

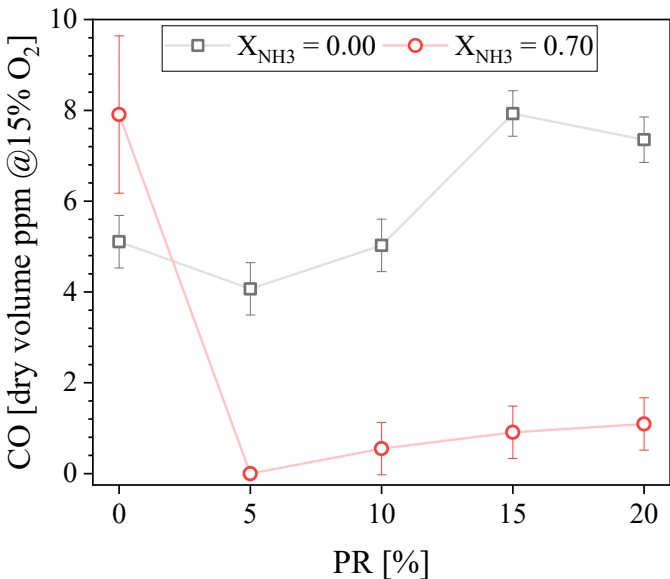

**Figure 5.** Measured CO concentration in the exhaust gases as a function of the PR.

Figure 6 presents the measured exhaust NO concentration (a) and NO percentage of increase (b) as a function of the PR. The NO concentration increases when PR increases for both fuel compositions examined. This trend can be explained by considering the key difference between the main flame and the pilot flame. While the main flame is a lean, technically-premixed flame, the pilot flame is non-premixed, which is prone to produce NO since the fuel burns on a stoichiometric contour, in a diffusion mode. This is true both if NO is produced via thermal or fuel pathways [17,25,26]. Therefore, increasing the PR increases the proportion of the fuel being burnt in a diffusion mode, leading to more NO.

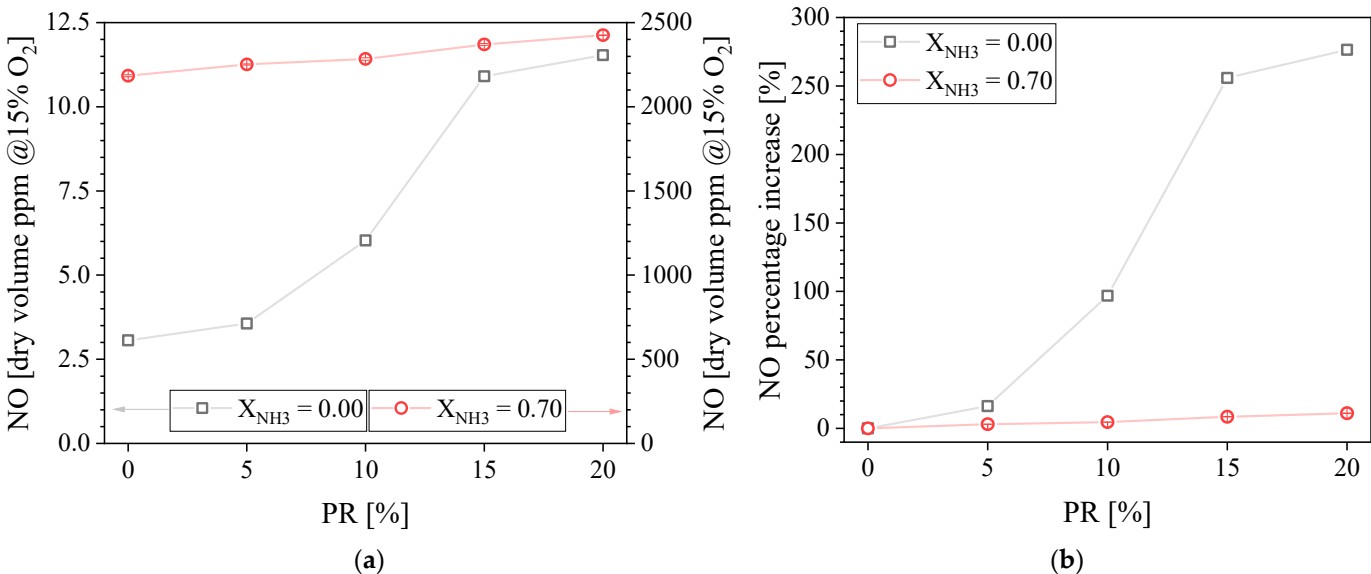

**Figure 6.** Measured NO concentration in the exhaust gases (**a**) and NO percentage of increase (**b**) as a function of the PR.

Although the NO emission increases with PR for both fuel compositions tested, it is important to look at the relative increment. Figure 6b presents the NO percentage of increase, with the NO concentration at PR = 0% taken as the baseline. It is found that increasing the PR more significantly affects the NO concentration percentage of increase for $CH_4$-air compared to $NH_3$-$CH_4$-air. The NO percentage of increases is 276% for $CH_4$–air flames when PR = 20%. It is only 11% for $NH_3$-$CH_4$-air flames when PR = 20%. The main reason is that $NO_x$ is produced via thermal-$NO_x$ pathways in the case of conventional hydrocarbons such as $CH_4$ [31,32], whereas it is mainly produced via fuel-$NO_x$ pathways for $NH_3$ combustion [16,33–35]. Indeed, the NO concentration is extremely low (~3 ppm) for the PR = 0 baseline for $CH_4$–air because the lean premixed main flame is not hot enough to generate a lot of NO. On the contrary, for the PR = 0 baseline for $NH_3$-$CH_4$-air, the lean premixed main flame already meets conditions to produce a large amount of $NO_x$ (~2300 ppm) via fuel-$NO_x$ pathways.

Nevertheless, Somarathne et al. [17] showed numerically that lower NO concentrations can be found for $NH_3$ non-premixed combustion at stoichiometric conditions compared to premixed combustion at lean equivalence ratios. This trend was experimentally confirmed by Khateeb et al. [25]. Therefore, increasing the PR would be expected to slightly reduce the NO emissions for $NH_3$-air flames, but this is not what we observed for $NH_3$–$CH_4$–air flames. The presence of $CH_4$ in the fuel blend may be responsible for the marginal NO concentration percentage of increase for $NH_3$–$CH_4$–air flames.

Figure 7 presents the $N_2O$ concentration in the exhaust gases. It is found to be constant at around 37 ppmvd for $CH_4$-air combustion regardless of the PR. In contrast, the $N_2O$ concentration decreases as the PR increases for $NH_3$–$CH_4$–air combustion. The maximum $N_2O$ concentration recorded is around 57 ppmdv at PR = 0% for $NH_3$–$CH_4$-air combustion. Okafor et al. [36] measured $N_2O$ emissions for $NH_3$-$CH_4$-air flames (up to $X_{NH3} \approx 0.52$) in an unpiloted, single-stage swirl combustor at atmospheric pressure. Consistent with our findings, they showed that increasing $X_{NH3}$ increases the $N_2O$ concentration for the lean equivalence ratio of $\phi = 0.6$. $N_2O$ is mainly produced through the reactions of NO with NH and $HO_2$ and is mostly consumed through reactions with H radicals and thermal dissociation. Takagi et al. [13] suggest that flames of $NH_3$ blended with hydrocarbons have a higher tendency to produce $NO_x$ than pure $NH_3$ flames because of the higher HCN concentration. HCN has a higher tendency to oxidize into NO through OH reactions than $NH_3$. In addition, Wargadalam et al. [14] demonstrated that NO and $N_2O$ formation is promoted by HCN oxidation. Therefore, the combination of the low temperature and low

H radicals concentration typical of lean combustion [36,37], the contribution of HCN to the formation of $N_2O$ in hydrocarbon-$NH_3$ flames [13,14], and the reaction quenching due to the unsteady behavior observed for this particular flame, can explain the higher $N_2O$ concentration measured at PR = 0%. It is important to note that the $N_2O$ concentration decreases as the PR increases, reaching almost the same concentration for the $NH_3$-$CH_4$ fuel blend at PR $\geq$ 15% compared with $CH_4$.

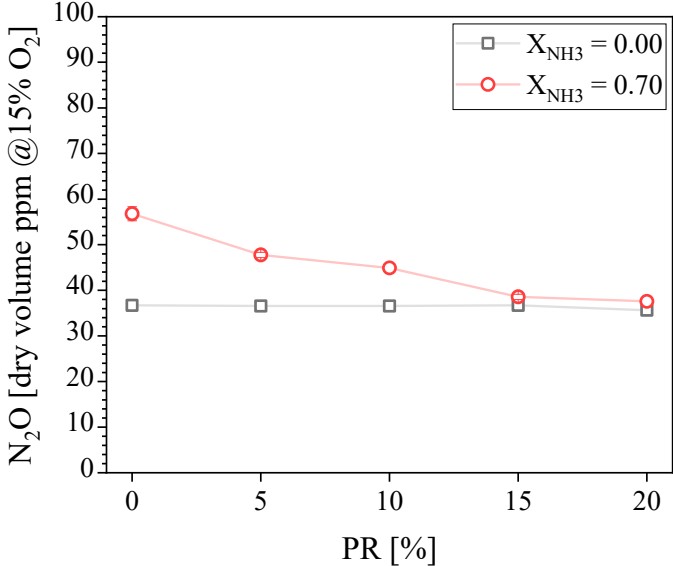

**Figure 7.** Measured $N_2O$ concentration in the exhaust gases as a function of the PR.

It is concluded that the presence of a pilot and its power have no influence on $NO_x$ concentration in the exhaust gases of this reduced-scale burner under the experimental conditions tested for $CH_4$. However, its absence leads to flame instabilities for $NH_3$-$CH_4$ fuel blends, in turn promoting $N_2O$ emissions.

### 3.2. Influence of the Pilot Fuel Composition (Pilot $X_{NH3}$)

#### 3.2.1. Flame Morphology

Figure 8 shows the time-averaged broadband images of $NH_3$–$CH_4$–air flames for three different PR (5, 10, and 15%) as a function of the pilot $X_{NH3}$. The main fuel composition was constant throughout this measurement series, with an $NH_3$ volume fraction of $X_{NH3}$ = 0.70. In general, the flames present the typical V-shape observed for swirling turbulent flames, and the orange-yellow hue is attributed to the $NH_2$ alpha band, $H_2O$ radicals [28,38], and $NO_2$* [29]. Interestingly, regardless of the pilot $X_{NH3}$, the flames with PR = 5% are more compact and anchored closer to the pilot liner outlet, compared to the flames with higher PR.

For a fixed PR, the time-averaged, main flame morphology does not appear to be significantly impacted by the variation of the pilot $X_{NH3}$. However, the structure and color immediately downstream of the pilot liner do change with pilot $X_{NH3}$. The region with the brighter orange-yellow hue issuing from the pilot becomes longer and increases in intensity when increasing the pilot $X_{NH3}$. In addition, increasing Pilot $X_{NH3}$ modifies slightly the color of the main flame, with hints of purple for the pure $CH_4$ pilot (Pilot $X_{NH3}$ = 0.0) to fully orange-yellow for the pure $NH_3$ pilot (Pilot $X_{NH3}$ = 1.0). This effect is most evident for PR = 15% where the contribution of the pilot is the largest. Note that, regardless of PR, resorting to a less reactive, pure $NH_3$ pilot did not appear to lead to significant flame instabilities.

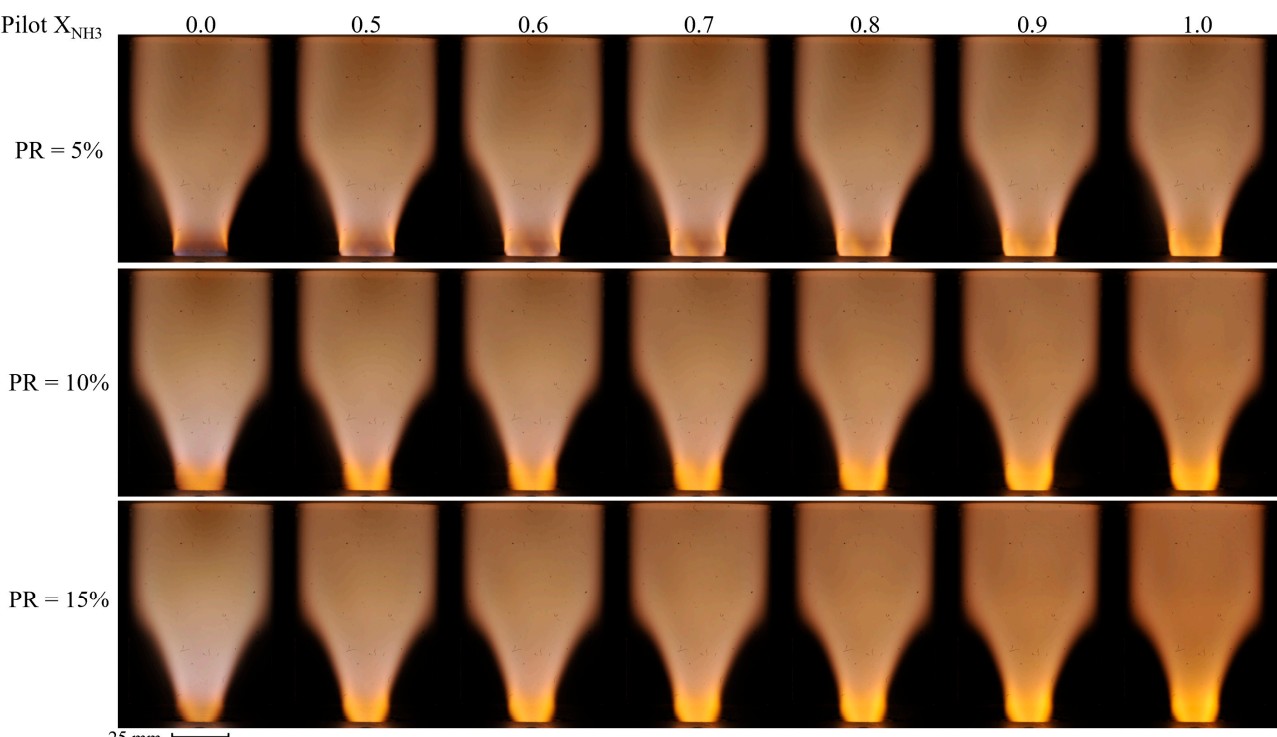

**Figure 8.** Time-averaged broadband images of $NH_3$–$CH_4$–air flames as a function of the pilot $NH_3$ fraction for PR = 5 (**top row**), 10 (**middle row**), and 15% (**bottom row**).

3.2.2. Exhaust Emissions

Figure 9 presents the CO concentrations in the exhaust gases as a function of the Pilot $X_{NH3}$ for three different PR values. All the conditions tested produced CO concentrations below 10 ppmvd. Increasing the $NH_3$ proportion in the pilot flame only increases the CO emissions above some threshold value; the CO concentration then increases rapidly. This threshold value depends on the PR; it is $X_{NH3}$ = 0.8 and 0.6 for PR = 10 and 15%, respectively. Although unburned fuel was not measured during this experimental campaign, it can be concluded, based on the CO concentration reported in Figure 9, that increasing Pilot $X_{NH3}$ eventually lead to a small drop in combustion efficiency.

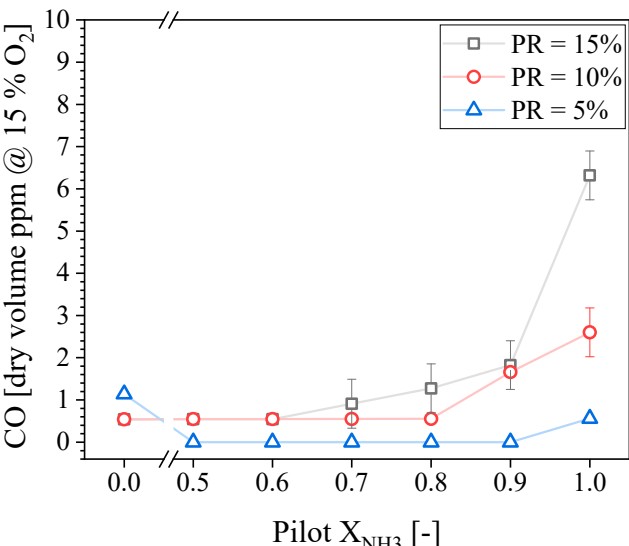

**Figure 9.** Measured CO concentration in the exhaust gases as a function of the pilot $NH_3$ for different PRs.

Figure 10 shows the NO concentration in the exhaust gases as a function of the Pilot $X_{NH3}$ for the three PR tested. The maximum and minimum NO concentrations recorded were around 2550 and 2100 ppmdv, and were both obtained for Pilot $X_{NH3}$ = 0.0 (pure $CH_4$), at PR = 15 and 5%, respectively. Conversely, the NO concentration is arguably the same for all PR values if Pilot $X_{NH3}$ = 1.0 (pure $NH_3$). Therefore, trends of NO with PR are found to be non-monotonic, and the directional sensitivity of NO with Pilot $X_{NH3}$ depends on PR. The NO concentration is not a strong function of the Pilot $X_{NH3}$ for the lower PR tested; it increases first until reaching its local maximum at Pilot $X_{NH3}$ = 0.80 and then it slowly decreases. In contrast, the NO concentration decreases monotonically as a function of the Pilot $X_{NH3}$ for PR = 10 and 15%, with a steeper reduction for the highest PR. These trends were further investigated through OH-PLIF, and the findings are now discussed.

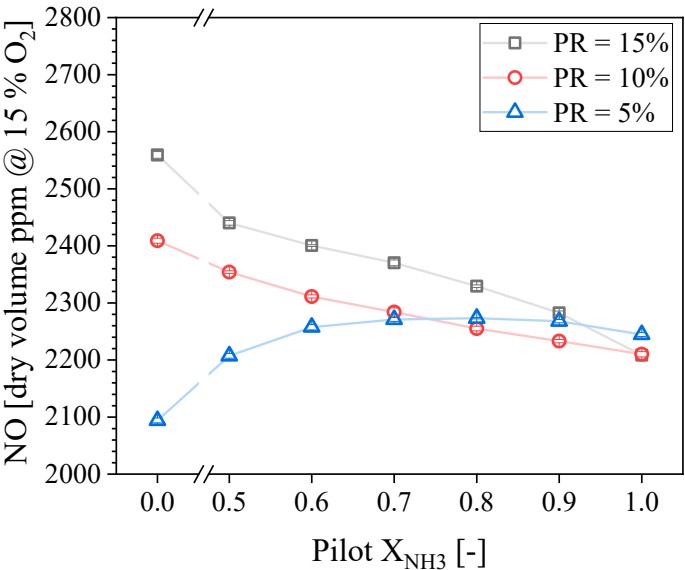

**Figure 10.** Measured NO concentration in the exhaust gases as a function of the pilot $NH_3$ for different PRs.

Recall that the fuel composition for the main flame was constant at $X_{NH3}$ = 0.70. Therefore, the change in NO concentration is due to the PR and Pilot $X_{NH3}$. It is known that the OH radical plays an important role on NO production in $NH_3$ flames [28,31,36]. Therefore, it is useful to study the influence of PR and Pilot $X_{NH3}$ on the concentration of OH radicals, which is performed here using OH-PLIF intensity as a proxy. Figure 11 presents the NO concentration in the exhaust gases as a function of the mean normalized OH intensity found in the interrogation area (see Figure 3) for the three PRs tested within the Pilot $X_{NH3}$ measurement series. Solid symbols correspond to Pilot $X_{NH3}$ = 0.0, and then the Pilot $X_{NH3}$ increases following the line for each PR. It is seen that modifying PR and/or Pilot $X_{NH3}$ affects the OH-PLIF intensity. Because the Q1(6) excitation line selected for these experiments has a moderate temperature sensitivity, it can be said that trends of OH-PLIF intensity are expected to match well that of the actual OH concentration. There is a good correlation between the OH-PLIF intensity and the NO concentration in the exhaust gases for PR = 15%. Increasing the Pilot $X_{NH3}$ results in lower OH concentration, hence, potentially explaining the lower NO exhaust concentration. Considering error bars, this positive correlation also exists for PR = 5%. However, a non-monotonic trend is found for PR = 10% and the linear correlation between NO and OH no longer holds. Figure 11 shows that the OH intensity for PR = 10% first decreases when Pilot $X_{NH3}$ decreases until = 0.6, and then it increases. This suggests that the OH concentration in the inner recirculation zone located immediately downstream of the pilot is not the only driver for the observed trends of NO concentration.

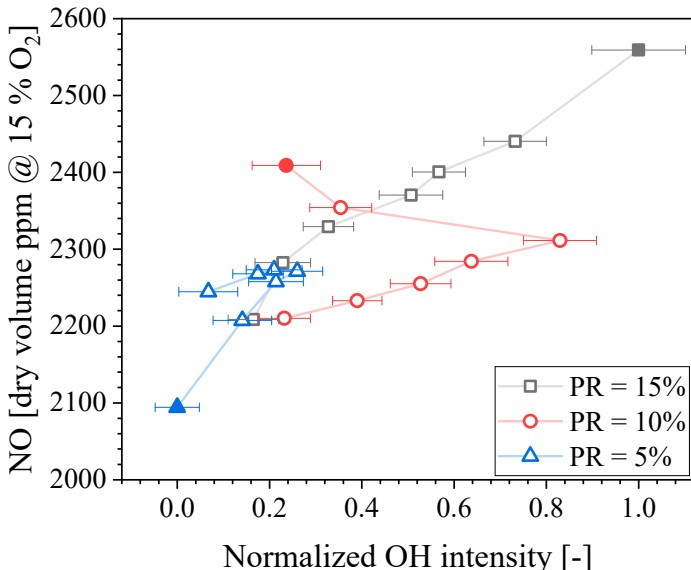

**Figure 11.** NO concentration in the exhaust gases as a function of the normalized OH intensity.

Figure 12 presents the normalized 2-D distributions of OH-PLIF probability measured for NH$_3$–CH$_4$–air flames. These distributions were obtained by taking an average of 600 binarized images for each condition. The threshold for binarization has been optimized to properly render the flame morphology, and isocontours (0.1, 0.3, 0.6, and 0.8 probabilities) have been overlaid to ease comparisons between conditions. Rows correspond to different PRs (5, 10, and 15%) and columns refer to different Pilot X$_{NH3}$. The horizontal white dashed line shows the laser sheet limit, located 1.5 mm above the pilot liner's outlet. In general, it is observed that both the PR and Pilot X$_{NH3}$ modify the flame morphology. Large differences can be found immediately downstream of the pilot liner, where hot products for both the main and pilot flames interact.

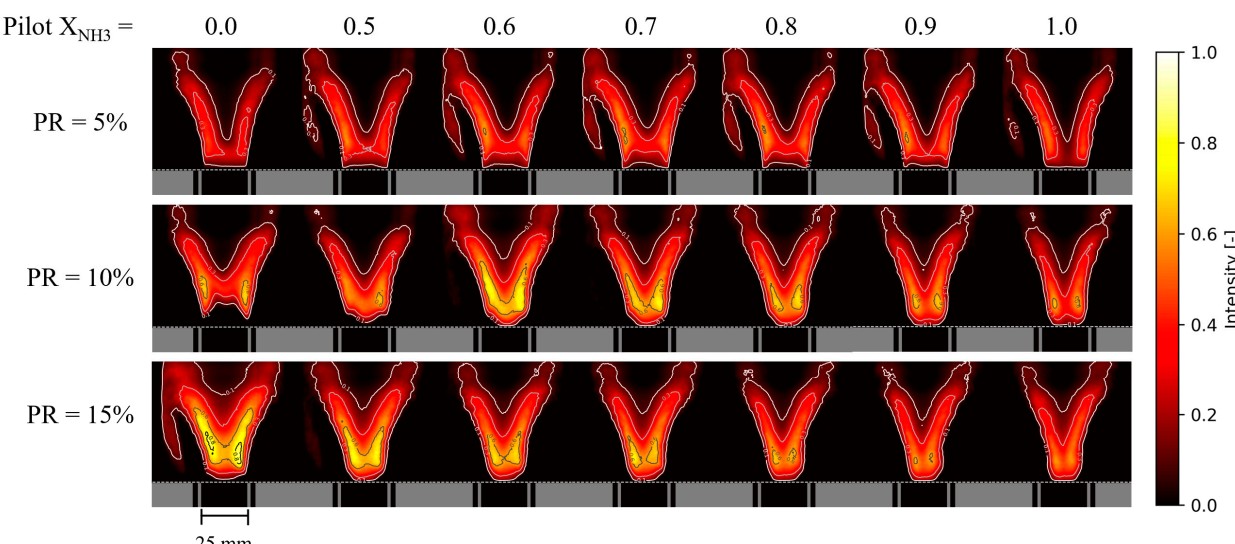

**Figure 12.** Normalized 2-D distributions of time-averaged OH intensity for NH$_3$–CH$_4$–air flames as a function of the pilot NH$_3$ fraction for PR = 5 (**top**), 10 (**middle**), and 15% (**bottom**).

The PR value for which the Pilot X$_{NH3}$ appears to influence the flame morphology most is PR = 10% and, here again, the trend is non-monotonic. For PR = 10% and Pilot X$_{NH3}$ = 0 the flame brush exhibits a comparatively large angle and the OH-PLIF signal is found to be quite weak/improbable immediately downstream of the pilot. As Pilot

$X_{NH3}$ increases up to 0.6, the presence of a strong OH-PLIF signal at this location becomes more apparent. However, when Pilot $X_{NH3}$ increases further up to 1.0, the OH-PLIF signal decays. This trend of flame morphology with Pilot $X_{NH3}$ exactly matches that of exhaust NO concentrations for PR = 10%. Although the complexity of this piloted swirled burner makes it difficult to relate flame morphology more accurately to global combustion properties such as NO emissions, a comparison of Figures 11 and 12 clearly shows that these properties are somewhat linked. In summary, our data show that both the OH concentration in the inner recirculation zone downstream of the pilot flame and the main flame's morphology drive NO emissions. In addition, it is clear that both OH concentration downstream of the pilot flame and the main flame's morphology is influenced by the pilot flame's properties.

Figure 13a presents the $N_2O$ concentration in the exhaust gases as a function of the pilot $X_{NH3}$ for different PRs. It is observed that the $N_2O$ trends are opposite to that of NO. The maximum and minimum $N_2O$ concentrations measured were 65 and 38 ppmdv for Pilot $X_{NH3}$ = 0.00, at PR = 5 and 15%, respectively. The $N_2O$ concentration slightly decreases as a function of the Pilot $X_{NH3}$ for PR = 5%, while it slightly increases for PR = 10%. In addition, the $N_2O$ concentration is almost constant at 38 ppmvd for PR = 15% except for Pilot $X_{NH3}$ = 1.0. Regardless of the exact trends, it can be said that the pilot flame's properties do not have a very large influence on $N_2O$ emissions.

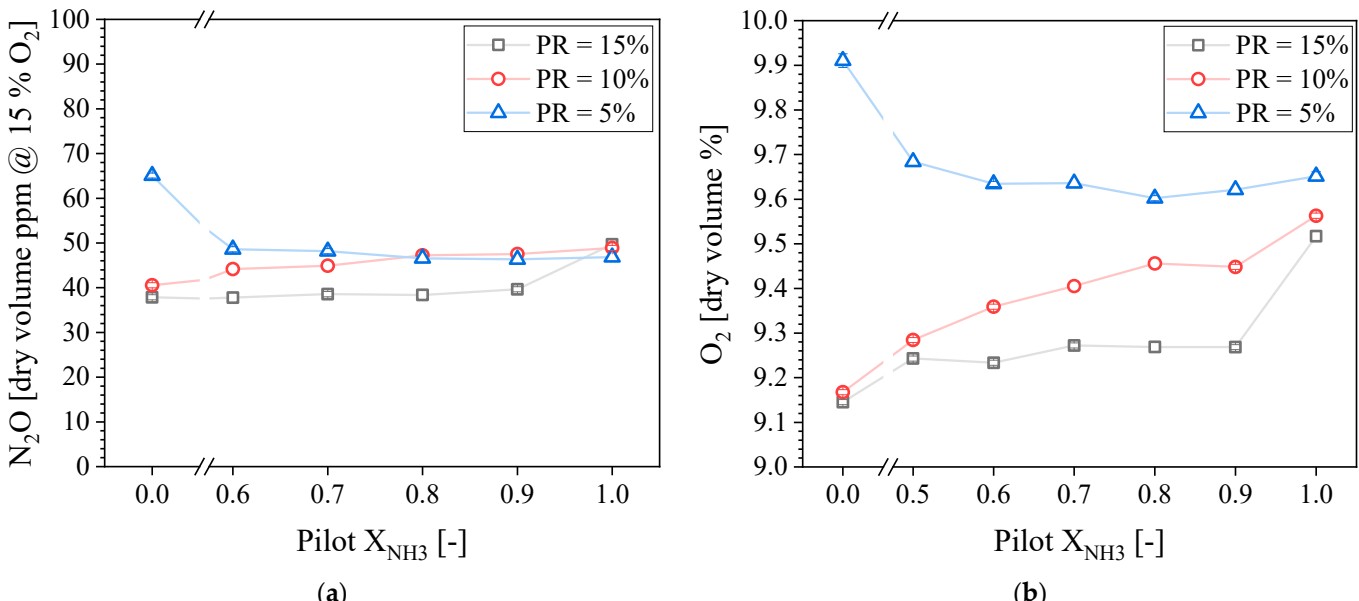

**Figure 13.** Measured $N_2O$ (**a**) and $O_2$ (**b**) concentration in the exhaust gases as a function of the pilot $NH_3$ for different PRs.

It is also interesting to observe the similarities in $N_2O$ and $O_2$ trends. Figure 13b presents the $O_2$ concentration in the exhaust gases. It is observed that trends of $O_2$ with Pilot $X_{NH3}$ and PR match that of $N_2O$. Higher $O_2$ concentrations found in the exhaust gases can be an indication of lower combustion efficiency and, in turn, lower flame temperature, which can be linked to an increase in $N_2O$. This is because $N_2O$ formed through the reaction zone is typically consumed through thermal dissociation.

## 4. Conclusions

This study experimentally assessed the impact of the pilot flame characteristics on the flame morphology and exhaust emissions of a piloted, reduced-scale burner fueled by either $CH_4$ or $NH_3$-$CH_4$ fuel blend. The pilot flame's power and fuel composition were varied, and flame imaging, CO, NO, and $N_2O$ molar fractions in the exhaust gases were measured. To further explain NO trends, OH-PLIF images were recorded for several conditions. The main findings are listed below:

- Both the pilot flame's power and fuel composition affect the flame morphology and exhaust emissions.
- A $CO_2$ concentration reduction of about 45% was achieved for $X_{NH3} = 0.70$ compared with pure $CH_4$, while still producing stable flames as long as $PR \geq 5\%$.
- Increasing the pilot flame's power increases the NO concentration in the exhaust for both $CH_4$ and $NH_3$-$CH_4$ flames, although the relative impact is largest for $CH_4$. This is because the main flame already produces a substantial amount of NO through fuel-$NO_x$ pathways for $NH_3$-$CH_4$, which is not the case for $CH_4$.
- Because boosting the pilot flame does not come with a large penalty on NO and $N_2O$ emissions for $NH_3$-$CH_4$ flames, its power can be increased more conveniently to ensure better flame stability than for $CH_4$ flames.
- The pilot flame's fuel composition has a rather complex influence on exhaust emissions. This is because it modifies both the flame morphology and the OH concentration in the inner recirculation zone.
- For a sufficiently powerful pilot ($PR > 5\%$), increasing the proportion of $NH_3$ in the pilot fuel decreases the NO concentration in the exhaust, which is arguably opposite to expectations. For those pilot power ratios, pure $CH_4$ pilot flames produced higher NO emissions. Conversely, the NO concentration was roughly constant for pure $NH_3$ pilot flames, regardless of the pilot power ratio.
- Data showed that the pilot power ratio and the pilot fuel composition modified the flame morphology and the OH concentration, which both influence NO emissions. While the flame morphology also plays a role, data showed that there is a strong positive correlation between the NO concentration in the exhaust and the OH concentration in the inner recirculation, which is consistent with the formation of NO through fuel-$NO_x$ pathways.

**Author Contributions:** Conceptualization, C.D.A.J., M.A.J. and T.F.G.; Data curation, C.D.A.J., S.C. and T.F.G.; Formal analysis, C.D.A.J., S.C., M.A.J. and T.F.G.; Funding acquisition, M.Y., A.J., T.F.G. and W.L.R.; Investigation, C.D.A.J. and S.C.; Methodology, C.D.A.J., M.A.J. and T.F.G.; Project administration, M.Y., A.J., T.F.G. and W.L.R.; Resources, M.Y., A.J., T.F.G. and W.L.R.; Software, C.D.A.J. and S.C.; Supervision, T.F.G. and W.L.R.; Visualization, C.D.A.J. and S.C.; Writing—original draft, C.D.A.J., S.C. and T.F.G.; Writing—review and editing, C.D.A.J., M.A.J., M.Y., A.J. and T.F.G. All authors have read and agreed to the published version of the manuscript.

**Funding:** The present work is supported by Saudi Aramco Research and Development Center under research agreement number RGC/3/3837-01-01 and by the Clean Combustion Research Center (CCRC) at King Abdullah University of Science and Technology (KAUST).

**Data Availability Statement:** The data presented in this study are available on request from the corresponding author.

**Acknowledgments:** Our thanks to Earnesto Thachil and Et-touhami Es-sebbar for their assistance with the OH-PLIF system setup.

**Conflicts of Interest:** The authors declare no conflict of interest.

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
