# Peer review of "Influence of the Pilot Flame on the Morphology and Exhaust Emissions of NH3-CH4-Air Swirl Flames Using a Reduced-Scale Burner at Atmospheric Pressure"

_energies, doi:10.3390/en16010231_

Round 1
Reviewer 1 Report
This manuscripts gives many information about CH4-NH3 blended fuel flame. Overall, it's well written, but the conclusion would be nice to reorganize if you can. The information contained in Absreact seems more informative than the information in Conclusion.
Author Response
The authors thank Reviewer 1 for the feedback. The conclusions have been expanded to better reflect the useful information contained in the abstract as follows (new text is underlined):
“• Both the pilot flame’s power and fuel composition affect the flame morphology and exhaust emissions.
- A CO2 concentration reduction of about 45% was achieved for XNH3 = 0.70 compared with pure CH4, while still producing stable flames as long as PR ≥ 5%.
- Increasing the pilot flame’s power increases the NO concentration in the exhaust for both CH4 and NH3-CH4 flames, although the relative impact is largest for CH4. This is because the main flame already produces a substantial amount of NO through fuel-NOx pathways for NH3-CH4, which is not the case for CH4.
- Because boosting the pilot flame does not come with a large penalty on NO and N2O emissions for NH3-CH4 flames, its power can be increased more conveniently to ensure better flame stability than for CH4 flames.
- The pilot flame’s fuel composition has a rather complex influence on exhaust emissions. This is because it modifies both the flame morphology and the OH concentration in the inner recirculation zone.
- For a sufficiently powerful pilot (PR > 5%), increasing the proportion of NH3 in the pilot fuel decreases the NO concentration in the exhaust, which is arguably opposite to expectations. For those pilot power ratios, pure CH4 pilot flames produced higher NO emissions. Conversely, the NO concentration was roughly constant for pure NH3 pilot flames, regardless of the pilot power ratio.
- Data showed that the pilot power ratio and the pilot fuel composition modified the flame morphology and the OH concentration, which both influence NO emissions. While the flame morphology also plays a role, data showed that there is a strong positive correlation between the NO concentration in the exhaust and the OH concentration in the inner recirculation, which is consistent with the formation of NO through fuel-NOx pathways.”
Reviewer 2 Report
The article presents an experimental study on the influence of the pilot flame characteristics on the flame morphology and exhaust emissions of a turbulent swirling flame. A reduced-scale burner, inspired by that fitted in the AE-T100 micro gas turbine, was employed as the experimental platform to evaluate methane (CH4) and an ammonia-methane fuel blend with an ammonia (NH3) volume fraction of 0.7.
The subject of the proposed article is very topical, both in theoretical and practical terms. Reducing exhaust emissions (by reducing knock combustion) is currently one of the most important problems faced by current vehicle engineering. The presented solution may be an important aspect in this matter.
The article consists of 4 chapters, presented in a logical order and a bibliography.
The introduction introduces us to the issue in detail, showing that it is a future-oriented issue and has a very large potential in modern engineering, which is also shown by the presented - current - bibliography.
The research methods do not raise any objections. They are presented in a legible and clear way.
The power ratio (PR) between the pilot flame and the main flame and the fuel composition of the pilot flame were varied. The pilot power ratio was varied from 0 to 20% for both fuel compositions tested. The NH3 volume fraction in the pilot flame ranged from pure CH4 to pure NH3 through various NH3–CH4 blends. Flame images and exhaust emissions, namely CO2, CO, NO, and N2O were recorded.
The results and conclusions were also presented clearly. The authors can expand their conclusions, but this is not something that significantly affects the quality of the article. It was found that increasing the pilot power ratio produces more stable flames and influences most of the exhaust emissions measured. The CO2 concentration in the exhaust gases was roughly constant for CH4-air or NH3–CH4–air flames. In addition, a CO2 concentration reduction of about 45% was achieved for XNH3 = 0.70 compared with pure CH4, while still producing stable flames as long as PR ≥ 5%. The pilot power ratio was found to have a higher relative impact on NO emissions for CH4 than for NH3–CH4, with measured exhaust NO percentage increments of about 276% and 11%, respectively. The N2O concentration was constant for all pilot power ratios for CH4 but it decreased when the pilot power ratio increased for NH3–CH4.
In my opinion, the article is suitable for publication in its current form.
Author Response
The authors thank Reviewer 2 for the feedback.
The only suggestion from Reviewer 2 is to expand the conclusion sections, which is also the only recommendation from Reviewer 1. We have improved our conclusion section, accordingly, as follows (new text is underlined):
“• Both the pilot flame’s power and fuel composition affect the flame morphology and exhaust emissions.
- A CO2 concentration reduction of about 45% was achieved for XNH3 = 0.70 compared with pure CH4, while still producing stable flames as long as PR ≥ 5%.
- Increasing the pilot flame’s power increases the NO concentration in the exhaust for both CH4 and NH3-CH4 flames, although the relative impact is largest for CH4. This is because the main flame already produces a substantial amount of NO through fuel-NOx pathways for NH3-CH4, which is not the case for CH4.
- Because boosting the pilot flame does not come with a large penalty on NO and N2O emissions for NH3-CH4 flames, its power can be increased more conveniently to ensure better flame stability than for CH4 flames.
- The pilot flame’s fuel composition has a rather complex influence on exhaust emissions. This is because it modifies both the flame morphology and the OH concentration in the inner recirculation zone.
- For a sufficiently powerful pilot (PR > 5%), increasing the proportion of NH3 in the pilot fuel decreases the NO concentration in the exhaust, which is arguably opposite to expectations. For those pilot power ratios, pure CH4 pilot flames produced higher NO emissions. Conversely, the NO concentration was roughly constant for pure NH3 pilot flames, regardless of the pilot power ratio.
- Data showed that the pilot power ratio and the pilot fuel composition modified the flame morphology and the OH concentration, which both influence NO emissions. While the flame morphology also plays a role, data showed that there is a strong positive correlation between the NO concentration in the exhaust and the OH concentration in the inner recirculation, which is consistent with the formation of NO through fuel-NOx pathways.”
Reviewer 3 Report
This manuscript is very weak from an academic contribution point of view. There is no comprehensive discussion about how chemical reactions are affected by stoichiometric conditions because changes in concentration, for example, then affect the flame. This work is very marginal. Moreover, incomplete accuracy during measurement. There is no detailed discussion of mass and energy conservation, and without touching energy/exergy analysis for example. This manuscript is a case study only, suitable for presentation at a conference. This manuscript does not provide any new findings.
Author Response
We appeal to the Editor to dismiss the comments from Reviewer 3. The poor, unconstructive feedback reflects a biased position towards either the subject or the research group or a lack of knowledge about the immediate research field. There is no proper justification provided, hence demeriting the purpose of the peer-reviewing process of this journal.
Reviewer 3 states that the work is very marginal because “There is no detailed discussion of mass and energy conservation, and without touching energy/exergy analysis, for example.” None of these types of analysis apply, or even make sense, in the context of this study. To illustrate, a search in Web of Science featuring the keywords “exergy,” “swirl,” and “flame” returns only 8 results (compared to many thousands for “swirl”, and “flame”). These 8 results all relate to the generation of entropy waves in combustors, which is a totally different research topic.
Regarding the statement “incomplete accuracy during measurement”: We provided accuracy and precision numbers for our instruments in our original manuscript as well as error bars in our plots, so we do not know what Reviewer 3 means. Again, because no justification or explanation is provided, we have no choice but to disregard this comment.
Indeed, we did not discuss the effects of stoichiometry on the flame. This is because these have been the focus of many previous studies, including multiple from our research group, and these have been well cited in our original manuscript (e.g., refs [16,18,21,25,26,28,36]).
Regarding the statement, “This manuscript does not provide any new findings”: Authors are not aware of any previous work on the effects of the pilot on premixed NH3-CH4-air swirling flames. Unfortunately, Reviewer 3 did not provide any example of other studies where our findings had already been reported.
